# Application of Knowledge-Driven Methods for Mineral Prospectivity Mapping of Polymetallic Sulfide Deposits in the Southwest Indian Ridge between 46° and 52°E

**Yao Ma [1], Jiangnan Zhao [1,\*], Yu Sui [2], Shili Liao [3]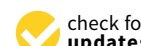 and Zongyao Zhang [1]**

1    School of Earth Resources, China University of Geosciences, Wuhan 430074, China;
     mayaoy2019@163.com (Y.M.); z18627012178@163.com (Z.Z.)
2    School of Resources and Environmental Science and Engineering, Hubei University of Science and
     Technology, Xianning 437100, China; jfysuiyu@163.com
3    Second Institute of Oceanography, State Oceanic Administration, Hangzhou 310012, China;
     liaosl.cug@gmail.com
\*    Correspondence: zhaojn@cug.edu.cn

**Abstract:** As a product of hydrothermal activity, seafloor polymetallic sulfide deposit has become the focus of marine mineral exploration due to its great prospects for mineralization potential. The mineral prospectivity mapping is a multiple process that involves weighting and integrating evidential layers to further explore the potential target areas, which can be categorized into data-driven and knowledge-driven methods. This paper describes the application of fuzzy logic and fuzzy analytic hierarchy process (AHP) models to process the data of the Southwest Indian Ocean Mid-Ridge seafloor sulfide deposit and delineate prospect areas. Nine spatial evidential layers representing the controlling factors for the formation and occurrence of polymetallic sulfide deposit were extracted to establish a prospecting prediction model. Fuzzy logic and fuzzy AHP models combine expert experience and fuzzy sets to assign weights to each layer and integrate the evidence layers to generate prospectivity map. Based on prediction-area (P-A) model, the optimal gamma operator ($\gamma$) values were determined to be 0.95 and 0.90 for fuzzy logic and fuzzy AHP to synthesize the evidence layers. The concentration-area (C-A) fractal method was used to classify different levels of metallogenic probability by determining corresponding thresholds. Finally, Receiver Operating Characteristic (ROC) curves were applied to measure the performance of the two prospectivity models. The results show that the areas under the ROC curve of the fuzzy logic and the fuzzy AHP model are 0.813 and 0.887, respectively, indicating that prediction based on knowledge-driven methods can effectively predict the metallogenic favorable area in the study area, opening the door for future exploration of seafloor polymetallic sulfide deposits.

**Keywords:** mineral prospectivity mapping; fuzzy logic; fuzzy analytical hierarchy process; seafloor polymetallic sulfide deposit; Southwest Indian Ridge

## 1. Introduction

With the progress of deep-sea exploration technology and depletion of terrestrial mineral resources, the exploration for seafloor mineral resources has been paid considerable attention by various countries [1]. Since the first discovery of seafloor hydrothermal vent at mid-ocean ridges in the late 1970s [2], the seafloor polymetallic sulfides have gradually become a new type of mineral resources rich in metallic elements of Cu, Fe, Zn, Mn, and Pb [3,4]. Due to its great potential value, it will

become an important part of exploitable marine mineral resource, and it is also one of the strategic alternative resources for sustainable development in the 21st century. In July 2011, the International Seabed Authority (ISA) approved China's application for a 10,000 km$^2$ region of the seafloor along the Southwest Indian Ridge (SWIR), and China have granted to exclusive exploration rights and preferential commercial mining rights in the area [5]. As regulated by the contract, by 2021 China can only retain exploration and mining rights of 25% of its area. In this regard, application of mineral prospectivity mapping (MPM) techniques to delineate potential areas that likely contain sulfide occurrences in this region of interest is urgent.

In the past decades, researches on MPM techniques have mainly focused on the following four aspects [6–8]: (1) MPM based on probability statistics (e.g., the weights-of-evidence); (2) MPM based on non-linear theory (e.g., fractal and multifractal); (3) MPM based on 3D modeling technology; and (4) MPM based on big data and machine learning (e.g., logistic regression, artificial neural network etc.). Generally, those MPM techniques can be divided into two categories of data-driven and knowledge-driven according to the types for assigning weights to each evidence layer [9,10]. The data-driven method analyzes and quantifies the spatial correlation between each evidential layer and the deposit location of common genesis, which is very applicable for well-explored areas [11]. Data-driven methods consist of logistic regression [12], weight of evidence [13], artificial neural network [14–16], support vector machine [6], random forest [17], and evidence belief function [18]. Incontrast, based on expert knowledge and judgment, the knowledge-driven approach qualitatively evaluates the relationship between each evidence layer and the deposits sought [13]. Analysts use expert opinions to evaluate the relative importance of spatial evidence and provide effective support for decision-making [19]. Common methods include fuzzy logic [20], index overlay [20], boolean logic [13] and fuzzy analytic hierarchy process (AHP) [19,21]. Comparing this approach to the data-driven method required a certain number of discovered mineral deposits; the knowledge-driven techniques are suitable for undeveloped areas where no or very few mineral deposits are known to occur. Due to a small number of hydrothermal points occurring in the study area, the knowledge-driven methods were selected for prediction in our research.

In this paper, fuzzy logic and fuzzy AHP were applied to quantitatively predict seafloor sulfide resources in the Southwest Indian Ridge between 46° and 52° E. The objectives include the demonstration of two knowledge-driven methods for MPM in seafloor sulfides, generation of seafloor sulfide prospectivity map for further exploration, and comparison of the performance of the two methods.

## 2. Geological Background

The Southwest Indian Ridge (SWIR), separating the Antarctic and African Plates, is the ultraslow spreading ridges with a full spreading rate of 14 km/Ma [22]. The west side of SWIR intersects with the Mid-Atlantic Ridge (MAR) and America-Antarctic Ridge (AAR) at the Bouvet Triple Junction (BTJ, 55° S, 00°40′ W), while the east side intersects with the Central Indian Ridge (CIR) and the Southeast Indian Ridge (SEIR) at the Rodrigues Triple Junction (RTJ, 25°30′ S, 70° E) (Figure 1), with a total length of about 8000 km [23]. Along-axis topographic and geophysical surveys revealed that the SWIR may be separated into some subsections, differing in the crustal thickness, obliquity of the ridge axis, topographic and geomorphological characteristics, mantle composition, and magmatic activity [1,24–26].

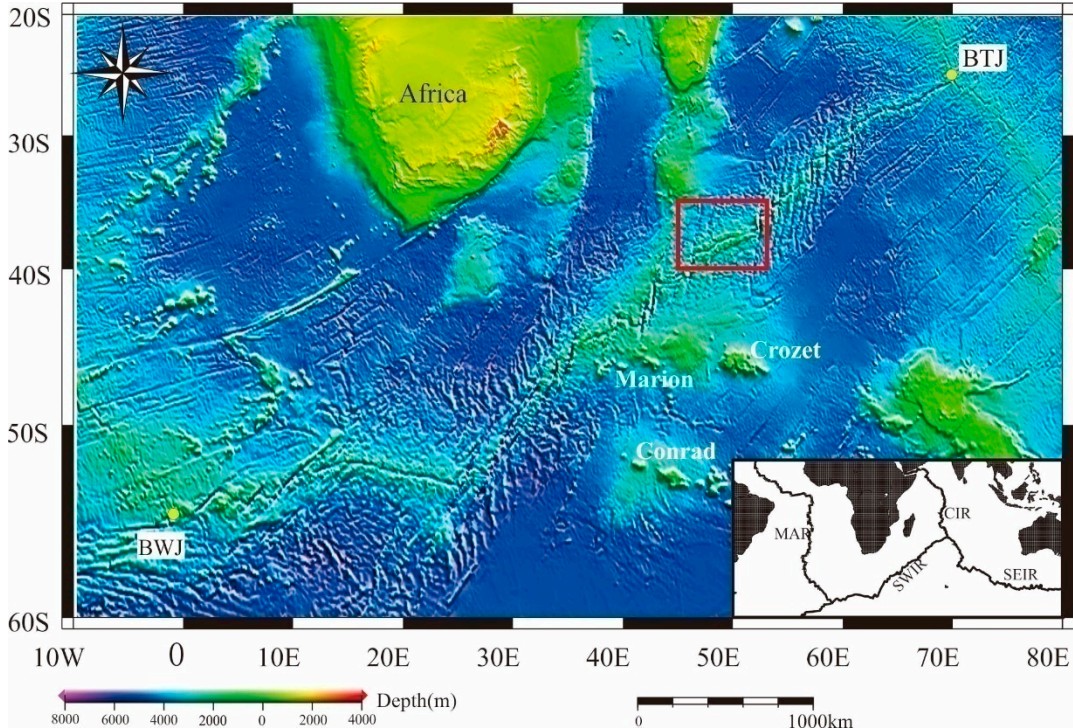

**Figure 1.** The study area in the central eastern part of the Southwest Indian Ridge (SWIR), ~900 km north of the Crozet hotspot, the red box indicates the range of the study area.

The study area belongs to the subsection between the Indomed and Gallieni transform faults and displays constant and slight overall obliquities, with a water depth varying from ~1500 m to ~4000 m [27]. The ridge axis in this region is E-NE striking with rugged terrain and well-developed central rift. The dominant rock in exposed seafloor is basalt, also including diabase, gabbro, and peridotite [25]. Previous studies have proved that this region has favorable conditions to generate seafloor hydrothermal activity and form sulfide deposits [26].It presents strong negative Bouguer gravity anomalies in this region, indicating active crust-mantle exchange to provide a heat source [26–28]. Melt inclusion and host glass compositions indicate higher degrees of melting fractions in the mantle melting column [29]. The detachment fault has exposed lower crust and mantle materials on the seafloor inferred by gravity, magnetic, borehole and seismic geophysical surveys, favoring the formation of an oceanic core complex and hydrothermal circulation system [30]. In addition, hydrothermal activity in this region is significantly associated with hot spots such as Crozet, Conrad, and Marion, indicating the strong ridge–hotspot interactions [23]. More and more countries and studies have paid increasing attention to this segment. The polymetallic sulfide exploration contract zone of China is also mainly located in this area. Since the Longqi hydrothermal field discovered in this region in 2007, some other hydrothermal fields have been discovered consecutively by Chinese Dayang cruises, such as Yuhuang, Duanqiao hydrothermal fields, etc. [1,28].

## 3. Data Acquisition and Method

### 3.1. Data

The study area is located in the southern hemisphere, with an area of about 350,000 km$^2$. Data sets of topographic, geophysical, geological, and other information are used as evidence sources for mineralization prediction. Topographic information mainly includes water depth and slope conditions. Geophysical information includes gravity and magnetic conditions. Geological information includes structure, ocean crust age, and sediment thickness. Other information mainly includes the distribution

characteristics of seismic points and the size of the spreading rates. The datum is derived from the public data of GeoMapApp. Water depth data and seismic point data are from the US Geological Survey. The former is a 30″resolutions, which are a fusion of multi-beam sounding and satellite altimetry. The latter selects data with magnitudes greater than 5 from 1950 to 2013. Gravity, magnetic data and spreading rate data are from the US National Geophysical Data Center, with a resolution of 2′. The structure is mainly derived from the inference of geophysical information. Ocean crust age data (resolution 2′) and sediment thickness data (resolution 5′) are from the National Oceanic and Atmospheric Administration. Hydrothermal point data in the study area mainly come from the hydrothermal vent database and published research results [1]. A spatial database was constructed using ArcGIS in accordance with Chinese database standards and specifications using geographic coordinate system GCS_WGS_1984.

### 3.2. Fuzzy Logic

In 1965, American mathematician L. Zadeh first proposed the fuzzy set theory, marking the birth of fuzzy mathematics [31].In order to establish a mathematical model of fuzzy objects, L. Zadeh generalized the concept of ordinary sets that only take two values of 0 and 1 as the concept of fuzzy sets that take infinitely many values in the interval [0,1], and used the concept of "membership" to accurately characterize the relationship between elements and fuzzy sets. The value of fuzzy membership can also be assigned through the fuzzification process [32]. After assigning fuzzy membership values to each category of dataset, the data set must be combined with one or more fuzzy factors to generate a map of mineral potential.

If $X$ is a collection of all evidence layers $X_i$ (i = 1, 2, 3,... $n$), each evidence layer has $r$ levels and is defined as ($j$ = 1, 2, 3,..., $r$), then the $n$ fuzzy sets $A_i$ (i = 1,2,3,..., $n$) of the evidence layer $X$ can be defined as

$$A_{ij} = \left\{ \left( x_{ij}, \mu_A \right) / x_{ij} \in X_i \right\}, \ (0 \le \mu_A \le 1) \tag{1}$$

where $\mu_A$ is a membership function. When $0.5 < \mu_A < 1$, $x_{ij}$ is favorable for mineralization. When $\mu_A = 0.5$, whether $x_{ij}$ is favorable for mineralization cannot be judged. When $0 < \mu_A < 0.5$, $x_{ij}$ is not favorable for mineralization. There are various forms of membership functions [33], and we uses the s-shaped membership function, which can be expressed as

$$\mu_A = \frac{1}{1 + e^{-a(x_{ij} - b)}} \tag{2}$$

In this formula, $x_{ij}$ calculated by $x_{ij} = w_i \times w_j$ represents the $j$-th level of the $i$-th evidence layer, $a$ and $b$ are determined by the shape of the function. Here, the value of $a$ and $b$ are assigned 0.1 and 50, respectively.

A fuzzy set operator is used to synthesize $A_i$ to generate a comprehensive fuzzy set $F$. Fuzzy AND, fuzzy OR, fuzzy algebraic sum, fuzzy algebraic product, and fuzzy gamma are the common fuzzy operators [34]. $F$ is the final score for each category of the evidence, which can be expressed as

$$F = \sum_{i=1}^{n} A_i \tag{3}$$

### 3.3. Fuzzy AHP

The concept of analytic hierarchy process was originally proposed by Satty in the mid-1970s, and is a well-known technique in the multiple criteria decision-making techniques [35,36]. The technology can address complex problems by converting them into simple forms [37,38]. In this paper, a fuzzy AHP method using triangular fuzzy number processing is used to establish a mineral potential prediction model. Seven steps involved in applying fuzzy AHP for MPM are summarized as follows [39,40]:

(1) Construction of a hierarchy. In the first step, the complex decision-making problem is reduced to a hierarchical structure of interrelated decision-making elements. Based on expert opinions, the evaluation system for mineral prediction is divided into three levels: the target level, the criterion level, and the plan level.

(2) Preparation of evidential layers. Map layers are converted into rasters in the GIS environment.

(3) Construction of pairwise comparison matrix. Pairwise comparisons can be obtained by asking experts about the association between the criterion and the target.

(4) Checking for consistency ratio (CR).

If the pairwise comparison matrix A = $(a_{ij})$ $m \times m$ satisfies $a_{ij} = a_{ik}a_{kj}$, I, j, k = 1,.., m, then A is considered to be completely consistent, otherwise it is considered to be inconsistent. A can be tested by the consistency ratio (CR) defined below. If CR ≤ 0.1, A is considered to be consistent; otherwise, the pairwise comparison matrix must be modified to satisfy the condition.

$$\text{CR} = \frac{(\lambda_{max})/(n-1)}{RI} \tag{4}$$

where $\lambda_{\max}$ is the maximum eigenvalue of the matrix, and RI is the average random consistency index. The consistency of the pairwise comparison judgments can not only measure the consistency of decision makers but also evaluates the quality of the model.

(5) Construction of fuzzy evaluation matrix.

The fuzzy comprehensiveness value of the *i*-th target is defined as

$$S_i = \sum_{j=1}^{M} M_{gi}^j \otimes \left[ \sum_{i=1}^{n} \sum_{j=1}^{m} M_{gi}^j \right]^{-1} \tag{5}$$

Among them, $M_{gi}^j$ (j = 1,2,..., m) is a triangular fuzzy number. If $M_1$ = ($l_1$, $m_1$, $\mu_1$) and $M_2$ = ($l_2$, $m_2$, $\mu_2$), then the probability of $M_1 > M_2$ is

$$V(M_2 > M_1) = \begin{cases} 1 & ,if\ m_2 \geq m_1 \\ 0 & ,if\ l_1 \geq u_2 \\ \frac{l_1 - l_2}{(m_2 - u_2) - (m_1 - l_1)} & ,otherwise \end{cases} \tag{6}$$

(6) Calculation of normalized weights.

(7) Using fuzzy operators.

The final vision map is generated after the data of all scheme layers and weights are synthesized using fuzzy operators.

## 4. Conceptual Model of Seafloor Polymetallic Sulfide Deposit

Seafloor polymetallic sulfide deposit is formed by water–rock hydrothermal processes in the oceanic crust, which are closely related to tectonic and magmatic activities and wall rock properties [41]. Many geological factors, such as water depth, stability of hydrothermal system, permeability, mixing process, fluid boiling, venting, and geological cap rock conditions, influence the metallogenic processes and control the occurrence of polymetallic sulfides [42,43]. As shown in the metallogenic model (Figure 2), the formation of Duanqiao hydrothermal field for polymetallic sulfides was largely due to the role of the normal faults with high angle as well as long-term and stable supply of magma and heat sources from axial magma chamber (AMC) [43].Hydrothermal circulation strongly influenced by magmatic and tectonic processes, controlled the transfer of energy and material [5].

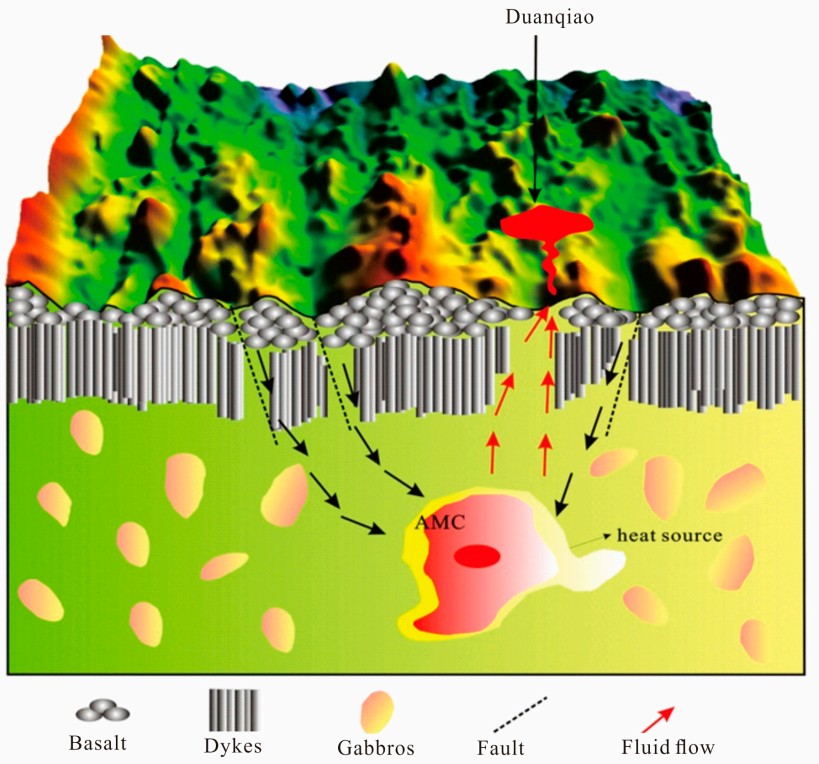

**Figure 2.** A metallogenic model of Duanqiao hydrothermal region in SWIR [43].

The mineral systems approach is usually proposed to the identify crucial criteria and establish a conceptual model, and its major components are as follows: (i) sources of mineralizing fluids and metallic element compositions; (ii) energy gradients that drive the movement of fluids; (iii) migration pathways for large amount of fluids; (iv) physical and chemical traps for ore precipitation; and finally (v) preservation of the deposits. In the seafloor sulfide mineral system of this research, three groups of factors are considered to be essential, which are source, pathway of ore-forming fluids, and favorable physico–chemical conditions for sulfide deposition and preservation. On the basis of the latest data of global hydrothermal vent sites and the results of seafloor polymetallic sulfide deposits in SWIR [1,5,23,44,45], the conceptual model that constrains the formation and distribution of seafloor sulfide deposits was established. The exploration criteria can be inferred from various datasets to extract direct or indirect criteria in the form of evidential layers.

(1) Hydrothermal sites. Hydrothermal vents and related sulfide samples are discovered by the near-bottom surveys, such as deep-sea tows, remotely operated underwater vehicles, human occupied vehicles, and autonomous underwater vehicles [1]. Previous works have also focused on characterizing hydrothermal plume patterns to explore new hydrothermal sites [46]. There will be a higher possibility to find sulfides near or in the vicinity of the hydrothermal locality compared to other areas. In this regard, hydrothermal sites directly provide material source information.

(2) Topography factor. Topography is closely related to the physico–chemical conditions. Hydrothermal activity in different tectonic environments has significantly different water depth distribution characteristics. Water depth plays a crucial role in the formation and distribution of polymetallic sulfides on the ocean floor. Studies have shown that most seafloor polymetallic sulfide occurrences occur at the water depth range between 2000 and 3000 m [47].Modern submarine hydrothermal activity is mainly distributed on negative terrain in high-bottom terrain, so slope can be used as a predictive factor.

(3) Geological factors. Geological factors are directly associated with the pathway of ore-forming fluids (e.g., Subsea fault) and physico–chemical conditions (e.g., the sedimentary cap). Subsea fault structure is the most important ore-conducting and ore-holding channel in the hydrothermal active area.

Submarine hydrothermal activity and its mineralization are closely related to the tectonic evolution of the oceanic plate. The distance between the position of the hydrothermal point and the ocean ridge can be limited by the age of the ocean crust. In the hydrothermal active zone, the presence of the sedimentary cap layer facilitates the precipitation and aggregation of metal elements, which can obviously promote the formation of hydrothermal products such as hydrothermal sulfides on a larger scale.

(4) Geophysical factor. Geophysical factors can also indirectly support the physico–chemical information. If the seafloor hydrothermal sulfide area has thick sedimentary layers and the thickness of the sedimentary layer changes greatly, the density difference will become larger. There is a certain magnetic difference between the sulfide deposit and the surrounding rocks, which can indicate the existence of the sulfide deposit. Therefore, gravity and magnetic anomalies are important prospecting indicators in the exploration of seabed polymetallic sulfide resources.

(5) Others. Seismic and volcanic activities on the sea floor mean that regional crustal activity is more active. Different spreading rates of the mid-ocean ridge tectonic environment have significantly different characteristics of deep magmatic activity, fault structure, and crustal thickness. They can indirectly reflect the material source and pathway of ore-forming fluids information.

These ore-controlling layers in the SWIR were used as evidence, which is summarized as follows: water depth, slope, gravity, magnetic force, structure, ocean crust age, sediment thickness, seismic point density, and spreading rate. The results of each layer are shown in Figure 3. Buffer analysis was conducted on the structural data using ArcGIS. The central buffer radius was 5000m, the buffer interval was set to 5000 m, and the maximum radius was 30,000 m. The dilation rate was analyzed by inverse distance weighting. Constructed data were reclassified based on buffer level; other data were reclassified with quantiles at intervals.

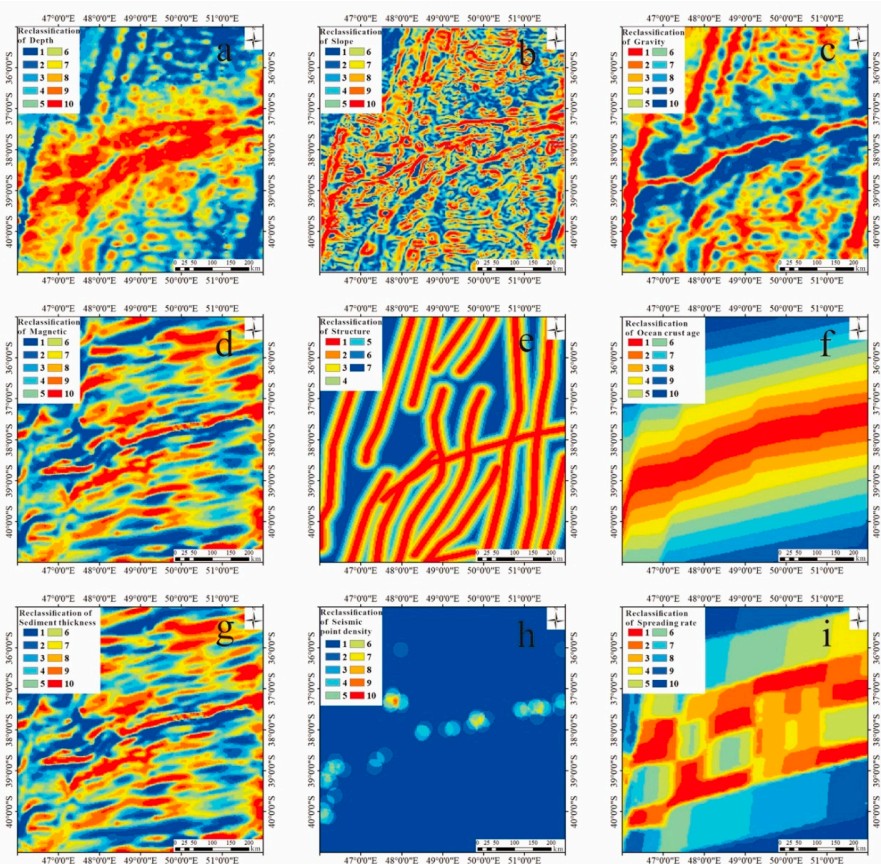

**Figure 3.** Reclassification of evidence layer.(**a**) Water depth; (**b**) Slope;(**c**) Gravity; (**d**) Magnetic; (**e**) Structure; (**f**) Ocean crust age; (**g**) Sediment thickness;(**h**) Seismic point; (**i**) Spreading rate.

## 5. Results

### 5.1. Fuzzy Logic

There are various forms of membership function. In this paper, the S-shaped membership function is used to transform the evidence level into the value between 0 and 1. Based on interviewing experts using a series of targeted questionnaires, three geoscientist experts from relevant fields were invited to grade each layer from 1 to 10. Comparing to results from only one expert, judgments derived from several experts are generally more accurate [48]. The experts considered the statistical characteristics of the data in comparison with their knowledge about each layer as evidence for the occurrence of seafloor polymetallic sulfide. As shown in Table 1, the average of each expert's score for each layer was taken as the final score. The greater the score, the more important the layers are to mineralization (Table 1). Then we processed the score of each level and used the information quantity method to explore the correlation between the classification of every evidential layer and the mineral occurrences. The value of the information amount is used for the weight allocation of each level. It assigns a larger weight value, and the fuzzy membership value vs. the level of each evidential layer can be calculated by Equation (2). In this paper, we only show the fuzzy membership of depth in Table 2.

**Table 1.** Final scores of 9 evidential layers.

| Evidence Layer | Expert A | Expert B | Expert C | Average Score |
|---|---|---|---|---|
| Depth | 9 | 8 | 9 | 8.67 |
| Slope | 6 | 6 | 6 | 6.00 |
| Gravity | 6 | 5 | 6 | 5.67 |
| Magnetic | 7 | 7 | 6 | 6.67 |
| Structure | 5 | 6 | 5 | 5.53 |
| Ocean crust age | 10 | 10 | 9 | 9.67 |
| Sediment thickness | 9 | 9 | 9 | 9.00 |
| Seismic points | 10 | 10 | 10 | 10.00 |
| Spreading rates | 8 | 8 | 8 | 8.00 |

**Table 2.** Fuzzy membership value of depth.

| CLASS | Grids | Points | Information Value | Weight of Layer | Weight of Class | Score of Class | Fuzzy Membership Value |
|---|---|---|---|---|---|---|---|
| −4968.42−−4044.88 | 6049 | 0 | 0 | 9 | 3 | 27 | 0.091122961 |
| −4044.88−−3777.19 | 6268 | 0 | 0 | 9 | 3 | 27 | 0.091122961 |
| −3777.19−−3603.19 | 6650 | 0 | 0 | 9 | 5 | 45 | 0.377540669 |
| −3603.19−−3442.58 | 6183 | 0 | 0 | 9 | 5 | 45 | 0.377540669 |
| −3442.58−−3281.96 | 6615 | 0 | 0 | 9 | 5 | 45 | 0.377540669 |
| −3281.96−−3107.96 | 6500 | 1 | 0.028724 | 9 | 7 | 63 | 0.785834983 |
| −3107.96−−2920.58 | 6268 | 3 | 0.52163 | 9 | 10 | 90 | 0.98201379 |
| −2920.58−−2679.65 | 6077 | 1 | 0.057948 | 9 | 8 | 72 | 0.900249511 |
| −2679.65−−2264.73 | 6040 | 2 | 0.361631 | 9 | 9 | 81 | 0.956892745 |
| −2264.73−−1555.35 | 5850 | 2 | 0.375512 | 9 | 9 | 81 | 0.956892745 |

### 5.2. Fuzzy AHP

In accordance with the steps of the fuzzy AHP, a hierarchical tree was first established according to the ore-controlling factors analysis of seafloor polymetallic sulfide deposits. The criterion layer consists of three sub-criteria, and the scheme layer consists of 9 sub-schemes (Figure 4). Three experts participated in constructing pairwise comparison matrix. Taking the criterion layer as an example, the pairwise comparison matrix constructed is given in Table 3. The final consistency ratio (CR) of all comparison matrices was less than 0.1 (Table 4). The fuzzy evaluation matrix is calculated using the trigonometric function fuzzy number. The results of the fuzzy evaluation matrix of the criterion layer are shown in Table 5. The weights of the nine layers of evidential layers are shown in Table 6. Then we

reclassify the reclassified data depending on the assigned weight values. Using the final weight values, multiplied results of the second reclassification is demonstrated in Table 7.

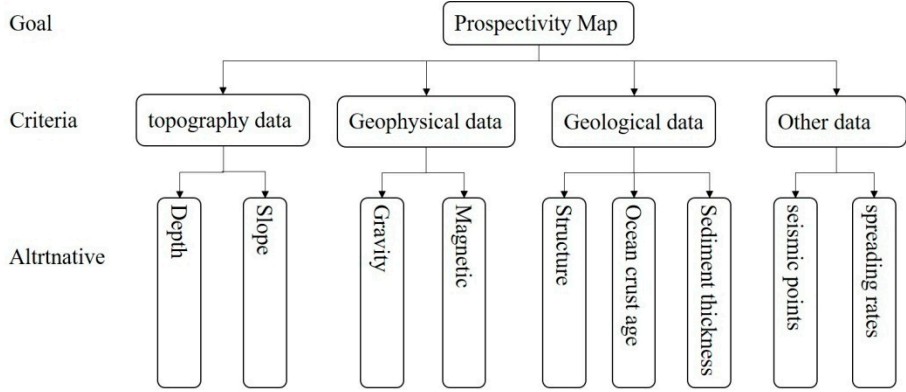

**Figure 4.** Hierarchical trees of this study.

**Table 3.** Pairwise comparison matrix with respect to criteria.

| Criterion | Topography Data | Geophysical Data | Geological | Other Data |
|---|---|---|---|---|
| Topography data | (1,1,1) | (2,1/2,1) | (1/2,1/3,1/2) | (1/2,1/3,1/4) |
| Geophysical data | (1/2,2,1) | (1,1,1) | (1/2,1/3,1/3) | (1/2,1/3,1/4) |
| Geological | (2,3,2) | (2,3,3) | (1,1,1) | (1,1/2,1/2) |
| Other data | (2,3,4) | (2,3,4) | (1,2,2) | (1,1,1) |

**Table 4.** Consistency ratio (CR) for pairwise comparison matrix.

| CI | Criteria | Topography Data | Geophysical Data | Geological | Other Data |
|---|---|---|---|---|---|
| DM1 | 0.0227 | 0 | 0 | 0.0515 | 0 |
| DM2 | 0.0454 | 0 | 0 | 0.0176 | 0 |
| DM3 | 0.0077 | 0 | 0 | 0.0707 | 0 |

**Table 5.** Fuzzy evaluation matrix with respect to criteria.

| Criterion | Topography Data | Geophysical Data | Geological | Other Data |
|---|---|---|---|---|
| Topography data | (1,1,1) | (0.5,1.1667,2) | (0.3333,0.4444,0.5) | (0.25,0.3611,0.5) |
| Geophysical data | (0.5,1.1667,2) | (1,1,1) | (0.3333,0.3889,0.5) | (0.25,0.3611,0.5) |
| Geological | (2,2.3333,3) | (2,2.6667,3) | (1,1,1) | (0.5,0.6667,1) |
| Other data | (2,3,4) | (2,3,4) | (1,1.6667,2) | (1,1,1) |

**Table 6.** Weights of criteria and alternatives.

| Criterion | Weight | Alternative | Weight | Final Weight |
|---|---|---|---|---|
| Topography data | 0.0557 | Depth | 0.6805 | 0.0379 |
| | | Slope | 0.3195 | 0.0178 |
| Geophysical data | 0.0552 | Gravity | 0.2988 | 0.0165 |
| | | Magnetic | 0.7012 | 0.0387 |
| Geological | 0.3821 | Structure | 0.1315 | 0.0502 |
| | | Ocean crust age | 0.3739 | 0.1429 |
| | | Sediment thickness | 0.4946 | 0.189 |
| Other data | 0.507 | Seismic points | 0.9881 | 0.501 |
| | | Spreading rates | 0.0119 | 0.006 |

**Table 7.** Reclassification of depth by assigned weight.

| Class | Grids | Points | Weight of Class |
|-------|-------|--------|-----------------|
| 1 | 0.379 | 3 | 10 |
| 2 | 0.3411 | 4 | 9 |
| 3 | 0.3032 | 1 | 8 |
| 4 | 0.2653 | 1 | 7 |
| 5 | 0.1895 | 0 | 5 |
| 6 | 0.1137 | 0 | 3 |

## 6. Discussion

### 6.1. Determination of Weight Value in Knowledge-Driven Model

We used two knowledge-driven methods for regional mineral resource prediction, and used expert knowledge and fuzzy set theory to provide a theoretical basis for calculating weights and handling the complexity of multiclass evidential layers [10]. In the fuzzy logic model, the weights between 0 and 1 were acquired by integrating the membership function with the score of evidential layers from each expert, and then they were assigned to each level of each layer. The weights of each layer do not add up to one. In the fuzzy AHP, experts first compare any two layers to form a pairwise comparison matrix, and then calculate the weights combining the triangular fuzzy numbers. The weights of each layer are in the range of 0–1, and the sum is equal to one. In the process of the pairwise comparison matrix construction, the consistency ratio of each factor after pairwise comparison by experts may be greater than 0.1, indicating inconsistency, which will affect the final prediction results. Therefore, knowledge-driven mineral resource prediction methods should consider these uncertain factors. Whether knowledge-driven method is appropriate depends on the nature of expert judgments that are applied in creating evidential maps and in integrating evidential maps. Accordingly, knowledge-driven MPM is also known as expert-driven MPM.

### 6.2. Generation of Seafloor Sulfide Prospectivity Map

Applying fuzzy analysis for mineral prospectivity mapping, a gamma operator ($\gamma$) is often used to integrate the nine evidence layers and generate prospectivity map. We use fuzzy gamma operator to perform fuzzy synthesis on all layers of evidence to ensure that the result of integration is within the range of maximum and minimum membership of the input variable. This can balance the "decreasing" and "increasing" effects of fuzzy algebraic sum and fuzzy algebraic product [49]. For determine the optimal $\gamma$ value, the prediction-area (P-A) analysis was used in our research [20]. As showed in Figure 5, in the fuzzy logic method, when $\gamma = 0.95$, the intersection point is at the highest position, which indicates that the predicted area at this time is relatively small and contains more known deposits. Using the same method (Figure 6), the optimal gamma value of the fuzzy AHP was determined to be 0.90.

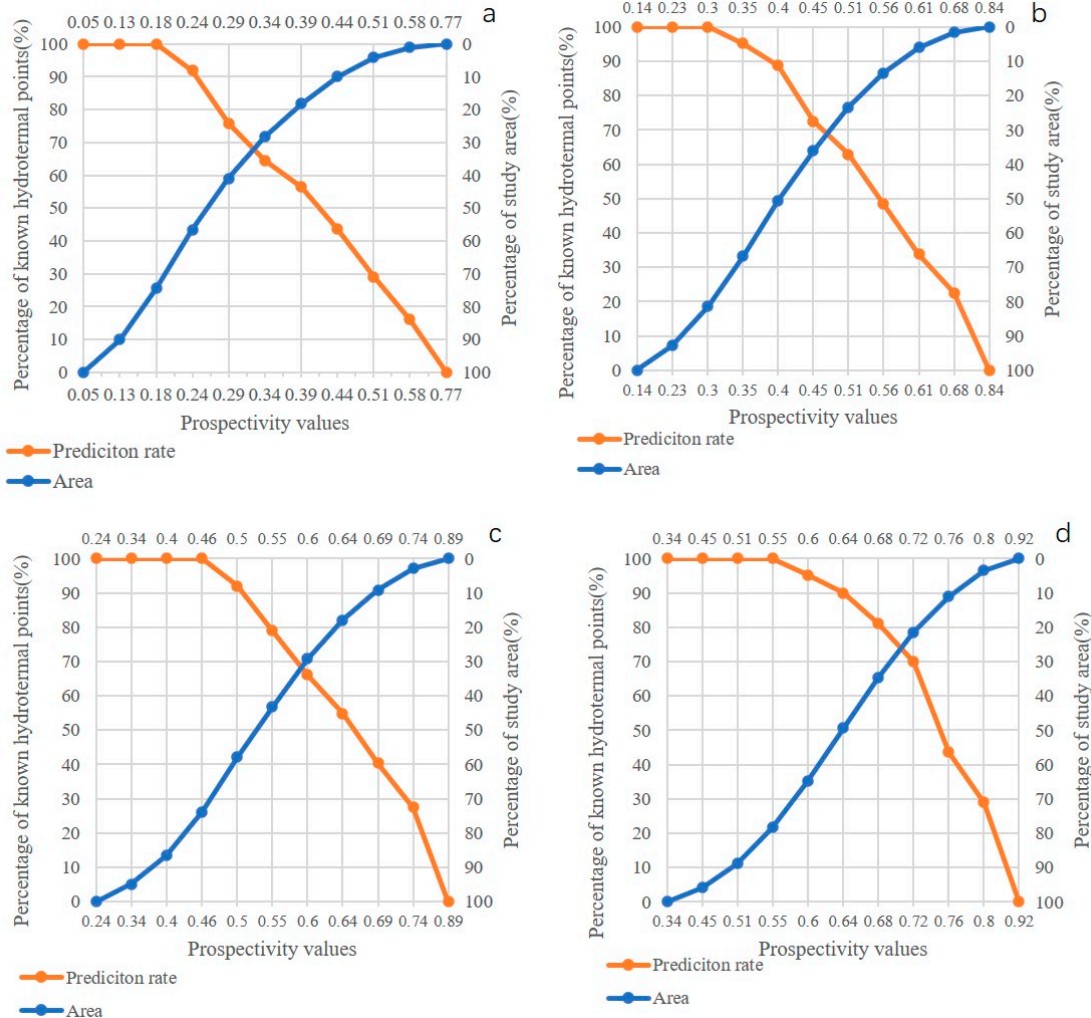

**Figure 5.** Comparison of different gamma values in fuzzy logic model. (**a**) $\gamma$ = 0.85, (**b**) $\gamma$ = 0.90, (**c**) $\gamma$ = 0.93, (**d**) $\gamma$ = 0.95.

The geological data processing software ArcFractal based on fractal/multifractal uses the concentration-area (C-A) fractal to divide the ore-forming prospect maps obtained by the two methods [50–52]. Based on the inflection point values analyzed, the division result of the C-A fractal models is shown in Figure 7, showing a power-law relationship between element concentration value and the area of cells. There were more than two enrichment steps, indicating the threshold values are 0.41 and 0.8 for fuzzy logic, and 0.31 and 0.44 for fuzzy AHP, which can be termed background, low and high favorable mineralization areas, respectively. The prospectivity maps showing the high favorable mineralization areas are illustrated in Figures 8 and 9, indicating that most prospect areas are distributed along the middle ridge, especially the intersection part with a series of N-S-striking transform faults.

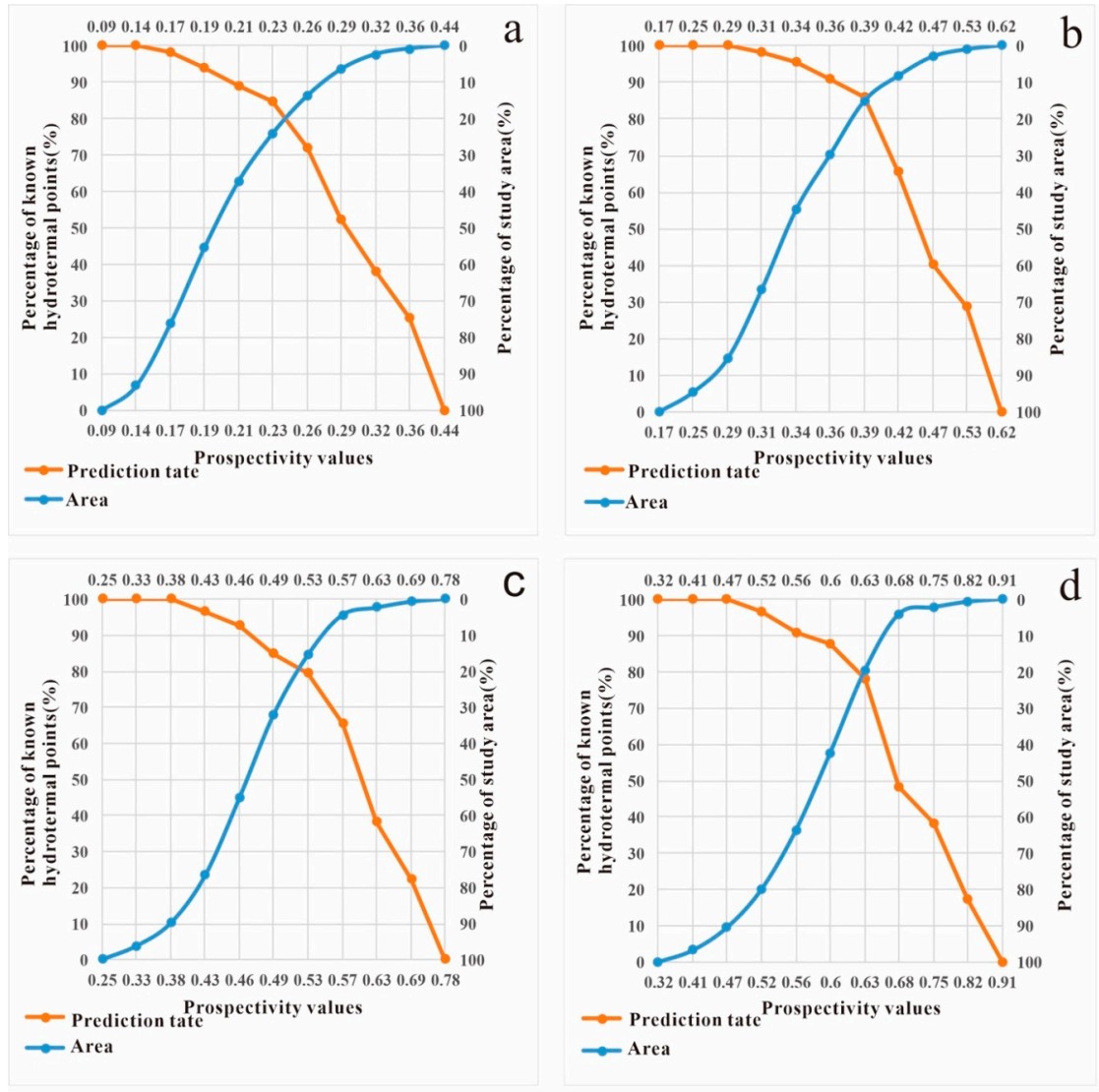

**Figure 6.** Comparison of different gamma values fuzzy AHP model. (**a**) $\gamma = 0.85$, (**b**) $\gamma = 0.90$, (**c**) $\gamma = 0.93$, (**d**) $\gamma = 0.95$.

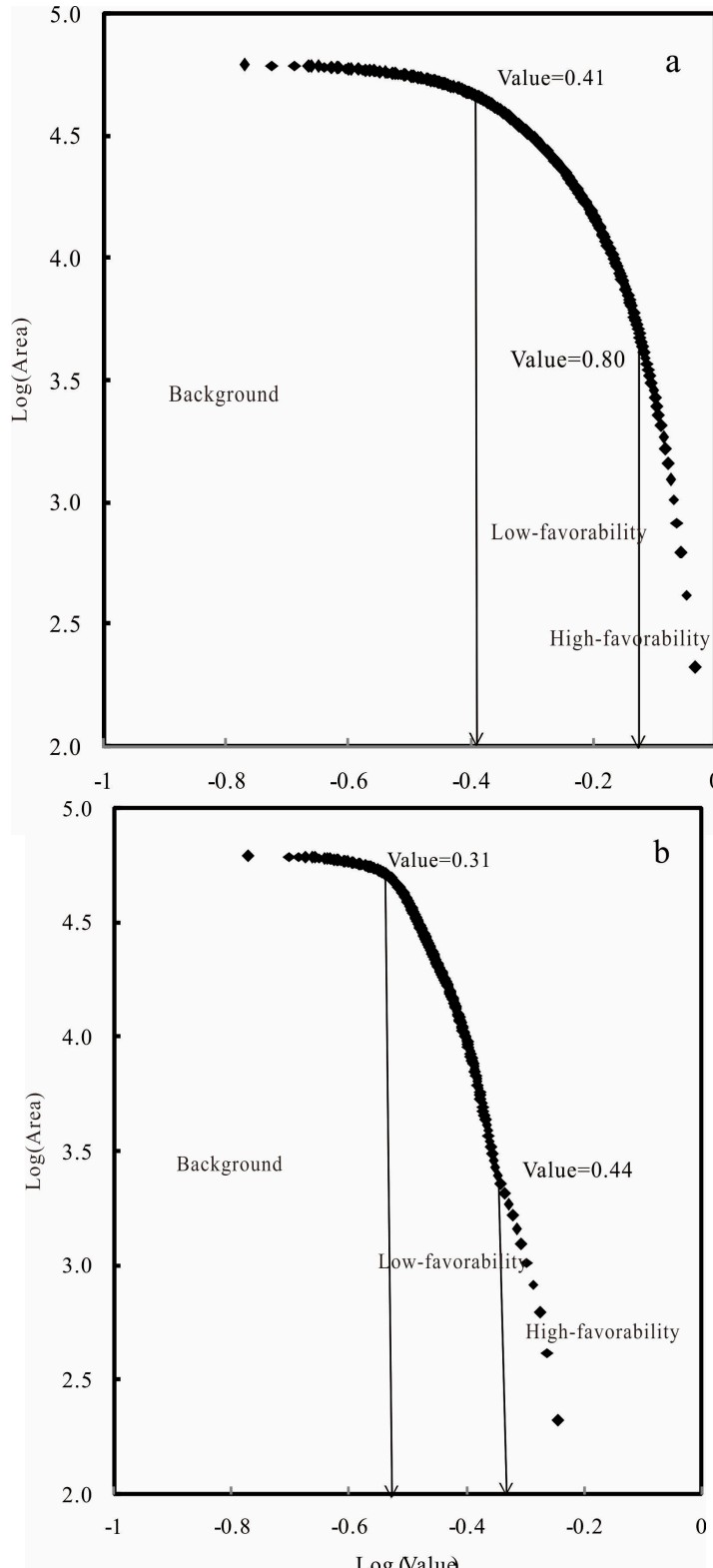

**Figure 7.** Concentration-area (C-A) model for the prospectivity map. (**a**) Based on fuzzy logic, (**b**) Based on fuzzy AHP. Value = 0.44.

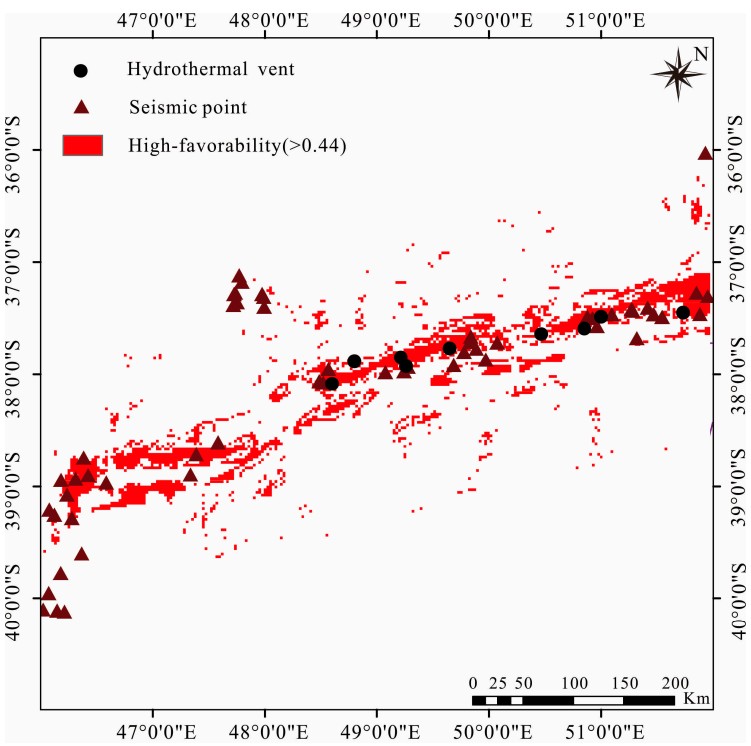

**Figure 8.** Mineral prospective map based on fuzzy analytical hierarchy process.

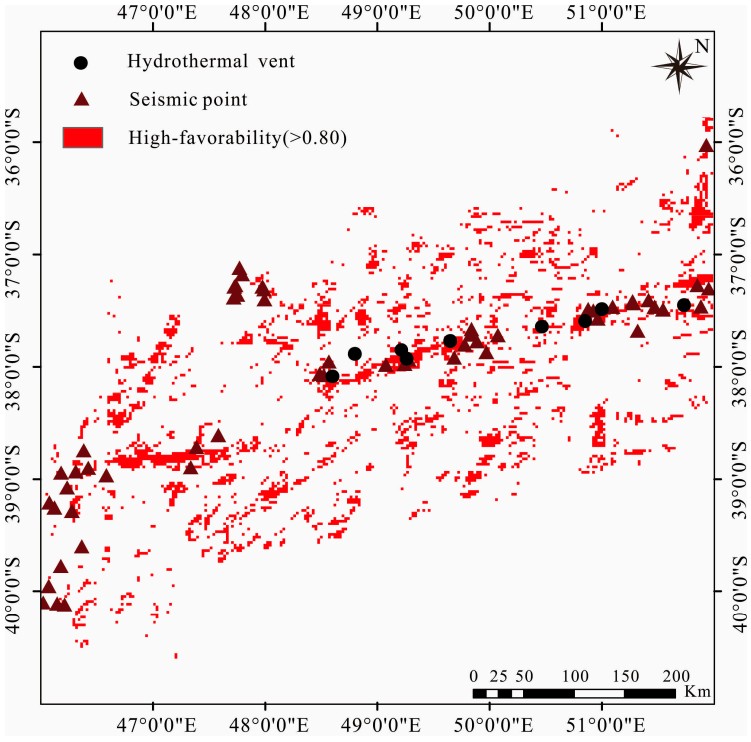

**Figure 9.** Mineral prospective map based on fuzzy logic.

### 6.3. Comparing the Performance of the Two Knowledge-Driven Methods

It is an important step towards any prospective modeling approach to validate predictive spatial models. The receiver operating characteristic (ROC) technique has been extensively applied to estimate the performance of mineral prospectivity models [53,54]. Finally, the ROC curve and statistical analysis

are used to assess the prediction effect of the method. Figure 10 presents the results of ROC analysis based on different mineral prediction probability thresholds. The area under the fuzzy logic curve (AUC) is 0.813, and the area under the fuzzy AHP curve (AUC) is 0.887; these show that the two models are capable of prospectivity mapping for seafloor sulfide. It also can be known that in the mineral prospect prediction based on fuzzy logic, the high mineralization favorable area includes 32 hydrothermal points and seismic points, and the area accounts for 11.28% of the entire research area (Table 8). In the prediction of mineral prospects based on the fuzzy AHP, there are 31 hydrothermal points and seismic points in the highly favorable area, and the area accounts for 9.37% of the entire research area. The fuzzy AHP and fuzzy logic method were applied to mineral prediction, and good results were achieved. The results show that the two knowledge-driven models are highly capable of mapping potential areas for sulfide mineralization, which opens up new avenues to prepare prospectivity maps. However, owing to the absence of certain exploration evidence data, such as hydrothermal alteration, geochemical anomaly, and hydrothermal plume deemed important in the prognosis of a sulfide deposit, the exploration criteria can be further facilitated using supplemental data sets to perfect this prospectivity model.

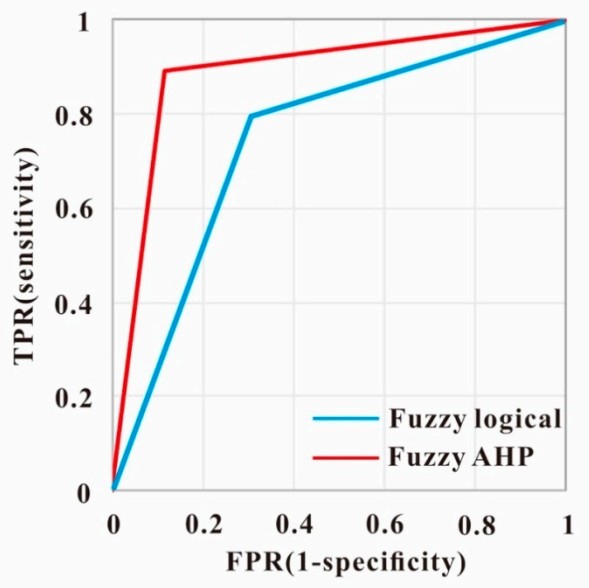

**Figure 10.** Verification of ROC curve.

**Table 8.** Statistical results of the two methods.

| Method | Ratio of Area of High-Favorability Regions to That of Study Area | Number of Hydrothermal Points and Seismic Points in High-Favorability Regions |
|---|---|---|
| Fuzzy logical | 11.28% | 32 |
| Fuzzy AHP | 9.37% | 31 |

## 7. Conclusions

This paper mainly discusses the application of knowledge-driven approaches to predict the seafloor polymetallic sulfides deposits in the mid-ridge of the Southwest Indian Ocean by integrating nine spatial evidential layers representing the controlling factors. The following conclusions were obtained. First, both fuzzy logic and fuzzy AHP can prioritize the prospect area effectively, which can transform the qualitative knowledge of experts into quantitative evaluation. Integrating fuzzy theory to obtain weight, those methods can overcome the shortage of binary response, and make expert opinions closer to reality. Second, a gamma operator can used to synthesize the evidence layer, the optimal $\gamma$ values were determined to be 0.95 and 0.9 for fuzzy logic and fuzzy AHP using

P-A plots, respectively. Third, The ROC curve and AUC were utilized to measure the performance of the prospectivity models, which show that the models are highly capable of mapping prospectivity because the AUC is greater than 0.5 and close to 1. The prospectivity mapping confirmed that there is significant potential for sulfide mineralization, which opens up new avenues for further exploration.

**Author Contributions:** Conceptualization, Y.M. and J.Z.; methodology, Y.M.; software, Y.M.; validation, Y.M., Y.S., and J.Z.; formal analysis, Y.M.; investigation, Y.M. and J.Z.; resources, Y.M.; data curation, Y.M., S.L., and Z.Z.; writing—original draft preparation, Y.M.; writing—review and editing, Y.M. and J.Z.; project administration, J.Z.; funding acquisition, J.Z. All authors have read and agreed to the published version of the manuscript.

**Funding:** This research was jointly supported by the National Key R&D Program (No. 2017YFC0306803), the Scientific Research Fund of Second Institute of Oceanography, MNR (SL2003), the Oceanic Interdisciplinary Program of Shanghai Jiao Tong University (SL2020MS031) and China Ocean's 13th Five-Year Project (DY135-S1-1-02).

**Acknowledgments:** We are also grateful for the reviewer's constructive comments and suggestions.

**Conflicts of Interest:** The authors declare no conflict of interest. No potential conflict of interest was reported by the authors.

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
