# Peer review of "Application of Knowledge-Driven Methods for Mineral Prospectivity Mapping of Polymetallic Sulfide Deposits in the Southwest Indian Ridge between 46° and 52°E"

_minerals, doi:10.3390/min10110970_

Round 1

Reviewer 1 Report

This revised version of the paper is a significant improvement and in particular I am pleased with the revisions to the section that articulates the exact MPM process utlitised by the authors. This is now much clearer and more understandable. 

The English grammar of the paper still needs work however, particularly the introductory sections which do not seem to have been improved much since the previous version. The paper needs a thorough review from an English language perspective. There are many examples of word running together without spaces and sentences that have not been properly constructed.

Author Response

This revised version of the paper is a significant improvement and in particular I am pleased with the revisions to the section that articulates the exact MPM process utlitised by the authors. This is now much clearer and more understandable.

Re: Thanks for your interest.

The English grammar of the paper still needs work however, particularly the introductory sections which do not seem to have been improved much since the previous version. The paper needs a thorough review from an English language perspective. There are many examples of word running together without spaces and sentences that have not been properly constructed.

Re: We also ask a native English research to polish the English.

Reviewer 2 Report

The article deals with an important research aspect of application importance. The estimation of polymetallic sulfide resources is needed at the stage of the deposit recognition and assessment of its economic significance.

The tools and testing methods used are properly selected. However, in the AHP method only three experts were used to prepare the parameters matrix. The AHP method should use more experts to rank data. The reliability and objectivity of the results is increased when more experts are used in the study. Whether it was a direct method of brainstorming experts type during a dedicated meeting or just a correspondence survey method. 

There are a lot of editorial mistakes in the manuscript text, mainly of the type of lack of space between words or use of capital letters.-The places of these defects I marked on the manuscript. 

Moreover, it requires a careful review of the citation of literature sources in the text of the article and in the list of the cited literature.I marked these points on the manuscript.

Author Response

The article deals with an important research aspect of application importance. The estimation of polymetallic sulfide resources is needed at the stage of the deposit recognition and assessment of its economic significance.

Re: Great suggestion. We focus on the delineation of prospective area that likely contains sulfide occurrences in this region of contract zone.

The tools and testing methods used are properly selected. However, in the AHP method only three experts were used to prepare the parameters matrix. The AHP method should use more experts to rank data. The reliability and objectivity of the results is increased when more experts are used in the study. Whether it was a direct method of brainstorming experts type during a dedicated meeting or just a correspondence survey method.

Re: Great suggestion. In our research, we select three experts to prepare the parameters matrix. The whole procedure is strict. We interview experts using a series of targeted questionnaires, three geoscientist experts are from relevant fields. They fully considered the statistical characteristics of the data in comparison with their knowledge about each layer as an evidence for the occurrence of seafloor polymetallic sulfide. So we think our results are credible and effective. We will consider to select more experts in application of knowledge-driven methods in further study.

There are a lot of editorial mistakes in the manuscript text, mainly of the type of lack of space between words or use of capital letters. The places of these defects I marked on the manuscript.

Moreover, it requires a careful review of the citation of literature sources in the text of the article and in the list of the cited literature. I marked these points on the manuscript.

Re: Revised. We also ask a native English research to polish the English.

This manuscript is a resubmission of an earlier submission. The following is a list of the peer review reports and author responses from that submission.

Round 1

Reviewer 1 Report

The paper has given interesting machine learning approach using latest fuzzy logic methods for mineral exploration. 

Introduction should have a better motivation paragraph - to highlight why this method is used. 

The literature review needs to improve and some latest papers in machine learning and mineral exploration needs to be included. A section - related work needs to be added with two paragraph that looks at mineral exploration and machine learning work done in past decade. These include:

  1. H Shirmard, E Farahbakhsh, AB Pour, AM Muslim, RD Müller, R Chandra, Integration of Selective Dimensionality Reduction Techniques for Mineral Exploration Using ASTER Satellite Data, Remote Sensing 12 (8), 1261, 2020
  2. E Farahbakhsh, R Chandra, HKH Olierook, R Scalzo, C Clark, SM Reddy, RD Muller, Computer vision-based framework for extracting geological lineaments from optical remote sensing data, International Journal of Remote Sensing, 1760-1787, 2020
  3. E Farahbakhsh, A Hezarkhani, T Eslamkish, A Bahroudi, R Chandra, Three-dimensional weights of evidence modeling of a deep-seated porphyry Cu deposit in Iran, https://arxiv.org/abs/1910.08162
  4. R Scalzo, D Kohn, H Olierook, G Houseman, R Chandra, M Girolami, S Cripps, Efficiency and robustness in Monte Carlo sampling of 3-D geophysical inversions with Obsidian v0. 1.2: Setting up for success, Geoscientific Model development, 2019

The methods section needs to increase to provide more details about the method. Fuzzy logic and advanced fuzzy logic method need to have better details, with some diagrams that link with the data that is used for this application. 

Results do not have standard deviation. Did you do multiple experiment runs of the model with different initialization? Why dont you have statistical measurements of error - with standard deviation and confidence interval in results?

Give details of how data was created into train and test split? How did you ensure you dont have overfitting?

Why fuzzy logic methods were applied and not neural networks or deep learning methods?

How do fuzzy logic compare with neural networks or other machine learning methods?

What are some of the directions for future research? Include this as a paragraph with discussion/conclusion section

Conclusion section should be rewritten to make it more presentable. 

Reviewer 2 Report

This paper presents an example of the application of knowledge-driven MPM methods to a particular case example, the SW Indian Ridge. Apart from the generally poor standard of the written English, I would make the following main criticisms:

  1. Any knowledge-driven MPM exercise will only ever be as good as the input geological conceptual framework. In the case of this paper, this framework is poorly presented and articulated. It appears that there has been only a cursory review of the relevant literature. In particular, there is no effort to discriminate between the setting of minor occurrences and larger ones that might be economic
  2. By their nature, knowledge-driven MPM processes (ie the details of the multiple steps in the process) are quite variable and subjective, because they are trying to fit the particular situation of interest and the constraints of the available data  to the available understanding (ie "knowledge") of the relative importance of factors. Therefore, it is essential in any publication of this type that the precise process is clearly articulated and the reason for deciding on the parameters at each stage clearly explained. This was not the case with this paper.
  3. A major part of this paper was comparing two different knowledge-driven MPM methods (Fuzzy Logic and Fuzzy Analytical Hierarchical Process or Fuzzy AHP) and making an assessment as to which was better. The authors conclude that the Fuzzy AHP is superior although the statistical differences between the two methods are minor and a visual inspection of the outputs (Figure 7 and 8) show no discernible difference. It is not a valid conclusion that the Fuzzy AHP method is better in my strong opinion; when all the uncertainties of these processes are considered, there appears to me to be no meaningful difference between them.
  4. As a practical matter, it is clear from Figures 7 and 8 that neither of these MPM process have actually been very effective in reducing the area of potential future exploration focus, with very large areas being defined as having "high favourability". I suspect that a simple map of major rift and transform structures would actually be better. Usually, when a MPM exercise produces this type of poor result, the reason is poor underlying geological input, as discussed in point 1 above.